# Video Prediction with Variational Temporal Hierarchies

## Abstract

Deep learning has shown promise for accurately predicting high-dimensional video sequences. Existing video prediction models succeeded in generating sharp but often short video sequences. Toward improving long-term video prediction, we study hierarchical latent variable models with levels that process at different time scales. To gain insights into the representations of such models, we study the information stored at each level of the hierarchy via the KL divergence, predictive entropy, datasets of varying speed, and generative distributions. Our analysis confirms that faster changing details are generally captured by lower levels, while slower changing facts are remembered by higher levels. On synthetic datasets where common methods fail after 25 frames, we show that temporally abstract latent variable models can make accurate predictions for up to 200 frames.

## 1 Introduction

Deep learning has enabled predicting video sequences from large datasets (Chiappa et al., 2017; Oh et al., 2015; Vondrick et al., 2016). For high-dimensional inputs such as video, there likely exists a more compact representation of the scene that facilitates long term prediction. Instead of learning dynamics in pixel space, latent dynamics models predict ahead in a more compact feature space (Doerr et al., 2018; Buesing et al., 2018; Karl et al., 2016; Hafner et al., 2019). This has the added benefit of increased computational efficiency and a lower memory footprint, allowing to predict thousands of sequences in parallel using a large batch size.

A lot of work in deep learning has focused on spatial abstraction, following the advent of convolutional networks (LeCun et al., 1989), such as the Variational Ladder Autoencoder (Zhao et al., 2017) that learns a hierarchy of features in images using networks of different capacities, along with playing an important role in the realm of video

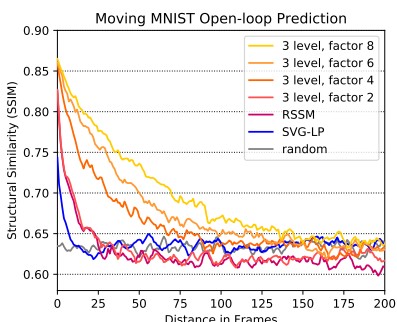

Figure 1: Mean SSIM over a test set of 100 sequences of open-loop prediction with Moving MNIST. All 3-level latent dynamics models, with temporal abstraction factors 2, 4, 6, and 8, have the same number of model parameters.

prediction models (Castrejón et al., 2019). Recent sequential models have incorporated temporal abstraction for learning dependencies in temporally distant observations (Koutník et al., 2014; Chung et al., 2016). Kim et al. (2019) proposed Variational Temporal Abstraction (VTA), in which they explored one level of temporal abstraction above the latent states, the transition of which was modeled using a Bernoulli random variable. In this paper, we intend to work in a more controlled setup than VTA for a qualitative and quantitative analysis of temporally abstract latent variable models.

In this paper, we study the benefits of temporal abstraction using a hierarchical latent dynamics model, trained using a variational objective. Each level in the hierarchy of this model temporally abstracts the level below by an adjustable factor. This model can perform long-horizon video prediction of 200 frames, while predicting accurate low-level information for a 6 times longer duration than the baseline model. We study the information stored at different levels of the hierarchy via KL divergence, predictive entropy, datasets of varying speeds, and generative distributions. In our experiments we show that this amounts to object location and identities for the Moving MNIST dataset, and the wall or floor patterns for the GQN mazes dataset (Eslami et al., 2018), stored at different levels.

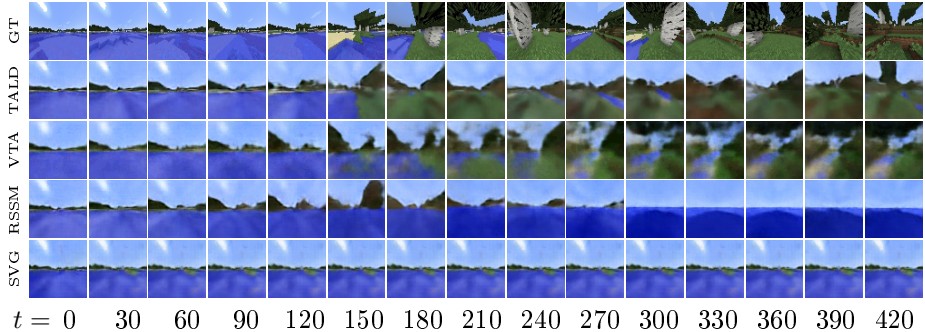

Figure 2: Long-horizon open-loop video prediction for the MineRL Navigate dataset (Guss et al., 2019), using our 3-level TALD model with temporal abstraction factor 6, compared with VTA, RSSM, and SVG-LP. We observe that TALD accurately predicts the movement of the scene from the ocean to the forest, and maintains that context until 420 frames. However, VTA predicts an implausible scene after 240 steps with blue and black skies in the same frame. RSSM predicts a plausible future as well where the player stays in the ocean as the distant forest moves out of the scene, whereas SVG-LP learns to copy the initial frame indefinitely and does not predict any new events in the future.

Our key contributions are summarized as follows:

- **Temporal Abstract Latent Dynamics (TALD)**   We introduce a simple model with different clock speeds at every level to study the properties of variational hierarchical dynamics.
- **Accurate long-term predictions**   Our form of temporal abstraction substantially improves for how long the model can accurately predict video frames into the future.
- **Adaptation to sequence speed**   We demonstrate that our model automatically adapts the amount of information processed at each level to the speed of the video sequence.
- **Separation of information**   We visualize the content represented at each level of the hierarchy to find location information in lower levels and object identity in higher levels.

## 2    RELATED WORK

**Generative video models**   A variety of methods have successfully approached video prediction using large datasets (Chiappa et al., 2017; Oh et al., 2015; Vondrick et al., 2016; Babaeizadeh et al., 2017; Gemici et al., 2017; Ha & Schmidhuber, 2018). Denton & Fergus (2018) proposed a stochastic video generation model with a learned prior that transitions in time, and is conditioned on past observations. Lee et al. (2018) proposed to use an adversarial loss with a variational latent variable model to produce naturalistic images, while Kumar et al. (2019) used flow-based generative modeling to directly optimize the likelihood of a video generation model. Recently, Weissenborn et al. (2020) scaled autoregressive models for video prediction using a three-dimensional self-attention mechanism and showed competitive results on real-world video datasets. On similar lines, Xu et al. (2018) proposed to use an entirely CNN-based architecture for modeling dependencies between sequential inputs.

**Latent dynamics models**   Latent dynamics models have evolved from latent space models that had access to low-dimensional features (Deisenroth & Rasmussen, 2011; Higuera et al., 2018), to models that can build a compact representation of visual scenes and facilitate video prediction purely in the latent space (Doerr et al., 2018; Buesing et al., 2018; Karl et al., 2016; Franceschi et al., 2020). The Variational RNN (Chung et al., 2015) uses an auto-regressive state transition that takes inputs from observations, making it computationally expensive to be used as an imagination module. Hafner et al. (2019) proposed a latent dynamics model, which is a combination of deterministic and stochastic states, that enables the model to deterministically remember all previous states and filter that information to obtain a distribution over the current state.

**Hierarchical latent variables**   Learning per-frame hierarchical structures has proven to be helpful in generating videos on long-term horizon (Wichers et al., 2018). Zhao et al. (2017) proposed the Variational Ladder Autoencoder (VLAE) that uses networks of different capacities at different levels of the hierarchy, encouraging the model to store high-level image features at the top level, and simple features at the bottom. Other recently proposed hierarchical models use a purely bottom-up inference approach with no interaction between the inference and generative models (Kingma & Welling,

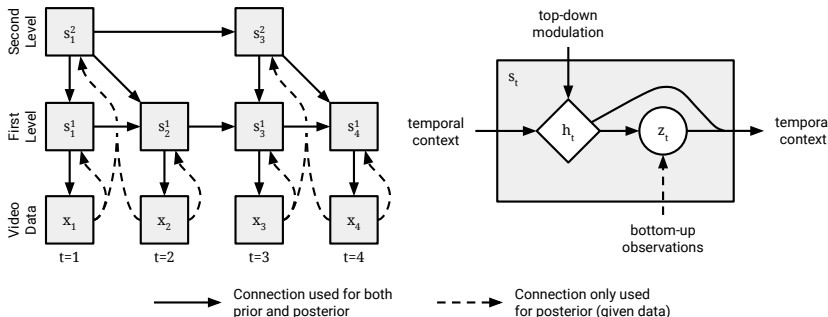

Figure 3: Temporally Abstract Latent Dynamics (TALD). Left is the structure of our recurrent model, in which each latent state $s_t^l$ in the second level abstracts two latent states in the first level. The solid arrows represent the generative model, while both solid and broken arrows comprise the inference model. On the right, we illustrate the internal components of the state variable, which comprises a deterministic state $h_t$ and a stochastic state $z_t$. The deterministic state processes all contextual information and passes it to the stochastic state to be used for either generation or inference.

2014; Rezende & Mohamed, 2015; Rezende et al., 2014). In contrast, Sønderby et al. (2016, LVAE) and Vahdat & Kautz (2020, NVAE) proposed to use a combination of bottom-up and top-down inference procedures, sharing parameters between the inference and generative distributions during the top-down pass. We incorporate this conditional structure in our model design as well.

**Temporal abstraction**  Identifying complex dependencies between temporally distant observations is a challenging task and has inspired a variety of fundamental work in recurrent models (Koutník et al., 2014; Chung et al., 2016). However, relatively few works have demonstrated modeling long-term dependencies using temporally abstract latent dynamics models (Wichers et al., 2018; Jaderberg et al., 2018). Recently, Kim et al. (2019) introduced Variational Temporal Abstraction (VTA) to learn temporally abstract latent spaces. They explored one level of temporal abstraction above the latent states, the transition of which was modeled using a Bernoulli random variable, that chose between 'copy' or 'update' steps. Inspired by this work, we aim to gain a deeper understanding of such temporally-abstract latent dynamics models. We perform our analysis on a model that is simplified to using fixed time scales for every level. Moreover, the lower level is a continuing chain in our model, whereas VTA resets transitions at a lower level when transitioning at a higher level.

## 3 Temporally Abstract Latent Dynamics

Long video sequences contain both information that is local to a few frames as well as global information that is shared among many frames. Traditional video prediction models that predict ahead at the frame rate of the video can struggle to retain information long enough to learn such long-term dependencies. We introduce Temporally Abstract Latent Dynamics (TALD) to learn long-term correlations of videos. Our model predicts ahead on multiple time scales to learn dependencies at different temporal levels, as visualized in Figure 3. We build our work upon the recurrent state-space model (RSSM; Hafner et al., 2019), the details of which can be found in Appendix A.

TALD consists of a hierarchy of recurrent latent variables, where each level transitions at a different clock speed. We slow down the transitions exponentially as we go up in the hierarchy, i.e. every level being slower than the level below by a factor of $k$. We denote a set of active timesteps for every level $l \in [1, L]$ as those steps in time where the state transition generates a new latent state,

$$\text{Active timesteps:} \qquad \mathcal{T}_l \doteq \{t \in [1, T] \mid t \bmod k^{l-1} = 1\}. \qquad (1)$$

At each level, we condition every window of $k$ latent states on a single latent variable in the level above. This can also be thought of as a hierarchy of latent variables where each level has the same clock speed, but performs a state transition every $k^{l-1}$ timesteps and copies the same state variable otherwise, so that $\forall t \notin \mathcal{T}_l$:

$$\text{Inactive states:} \qquad s_t^l \doteq s_{\max_\tau \{\tau \in \mathcal{T}_l \mid \tau \leq t\}}^l. \qquad (2)$$

**Joint distribution**  We can factorize the joint distribution of a sequence of observations and (active) latents at every level into two terms: (1) a decoder term conditioned on the latent states in the lowest

level, and (2) state transitions at all levels conditioned on the latent state of the last active timestep at the current level and the level above,

$$p(x_{1:T}, s_{1:T}^{1:L}) \doteq \left( \prod_{t=1}^{T} p(x_t \mid s_t^1) \right) \left( \prod_{l=1}^{L} \prod_{t \in \mathcal{T}_l} p(s_t^l \mid s_{t-1}^l, s_t^{l+1}) \right). \tag{3}$$

**Inference** For inference, TALD embeds observed frames using a CNN. A hierarchical recurrent network then summarizes the input embeddings, for which each (active) latent state at a level $l$ receives embeddings from $k^{l-1}$ observation frames (dashed lines in Figure 3). The latent state at the previous timestep at the current level, and the state belief at the level above also condition the posterior belief (solid lines in Figure 3). The input embeddings combined with this top-down and temporal context together condition the posterior belief $q_t^l$ over the latent state.

**Generation** The prior transition $p_t^l$ is computed by conditioning over the latent state at the previous timestep at the current level, and the state belief at the level above (solid lines in Figure 3).

**Decoding** Finally, the state beliefs at the bottom-most level are decoded using a transposed CNN to provide a training signal. To summarize, we utilize the following components in our model, $\forall\, l \in [1, L], t \in \mathcal{T}_l,$

$$\begin{aligned} &\text{Encoder:} & &e_t^l = e(x_{t:t+k^{l-1}-1}) \\ &\text{Posterior transition } q_t^l: & &q(s_t^l \mid s_{t-1}^l, s_t^{l+1}, e_t^l) \\ &\text{Prior transition } p_t^l: & &p(s_t^l \mid s_{t-1}^l, s_t^{l+1}) \\ &\text{Decoder:} & &p(x_t \mid s_t^1). \end{aligned} \tag{4}$$

**Training objective** Since we cannot compute the likelihood of the training data under the model in closed form, we use the ELBO as our training objective. This training objective optimizes a reconstruction loss at the lowest level, and a KL regularizer at every level in the hierarchy summed across active timesteps,

$$\max_{e,h,q,p} \sum_{t=1}^{T} \mathrm{E}_{q_t^1}[\ln p(x_t \mid s_t^1)] - \sum_{l=1}^{L} \sum_{t \in T_l} \mathrm{KL}[q_t^l \parallel p_t^l]. \tag{5}$$

The KL regularizer at each level limits the amount of information that filters through the encoder and stays in the posterior at that level. This encourages the model to utilize the state transitions and context from the level above as much as possible. Since the number of active timesteps decreases as we go higher in the hierarchy, the number of KL terms per level decreases as well. Hence it is easier for the model to push global information high up in the hierarchy and pay lesser KL penalty, instead of transitioning those bits with an identity transformation at a lower level.

**Stochastic and Deterministic Path** As illustrated in Figure 3 (right), we split the state $s_t^l$ into stochastic ($z_t^l$) and deterministic ($h_t^l$) parts (Hafner et al., 2019). The deterministic state is computed using the top-down and temporal context, which then conditions the stochastic state at that level.

The stochastic states follow a diagonal Gaussian, with mean and variance predicted by a neural network. We use a GRU (Cho et al., 2014) per level to update the deterministic state at every active timestep. All components in Equation 4 are trained jointly by optimizing Equation 5 using stochastic backpropagation with reparameterized sampling. Please refer to Appendix B for architectures details.

## 4 EXPERIMENTS

We aim to evaluate temporally-abstract latent dynamics models at modeling long-term dependencies in video. Moreover, we aim to understand how they separate information into different levels of the hierarchy. To investigate these questions, we train TALD described in Section 3, the temporally-abstract VTA model (Kim et al., 2019), the RSSM model without temporal abstraction (Hafner et al., 2019), and the image-space video prediction model SVG-LP (Denton & Fergus, 2018) on four datasets of varying complexity. We consider the well-established Moving MINST dataset (Srivastava et al., 2015), the KTH Action dataset (Schuldt et al.,

Table 1: KL divergence (in bits) between the posterior and prior at different levels of the hierarchy (summed in time). We observe that the amount of information at the (lowest) level 1 decreases as the model gets deeper.

| TALD | LEVEL 1 | LEVEL 2 | LEVEL 3 | LEVEL 4 |
|---|---|---|---|---|
| 4 LEVELS | 572.89 | 59.44 | 3.84 | 1.76e−4 |
| 3 LEVELS | 529.04 | 65.33 | 9.07 | - |
| 2 LEVELS | 561.79 | 56.52 | - | - |
| 1 LEVEL | 635.51 | - | - | - |

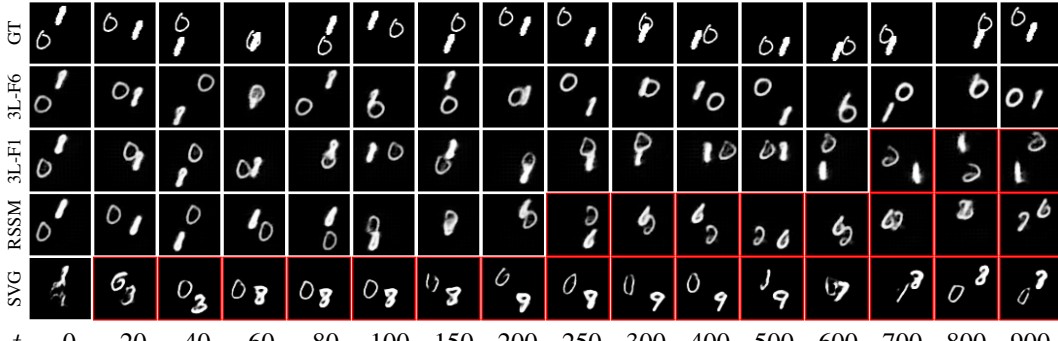

Figure 4: Long-horizon open-loop prediction for Moving MNIST. (L for levels, F for abstraction factor.) We illustrate samples from our TALD model with 3 levels and temporal abstraction factors: 6 (3L-F6), and 1 (3L-F1) (i.e. no temporal abstraction). We compare those with samples from the RSSM and SVG-LP baselines. Red boxes show instances in time from where models lose accurate object identity. We observe that TALD, with abstraction factor 6, is able to maintain accurate long-term dependencies in the form of object identities for 900 frames into the future.

2004), the GQN mazes dataset (Eslami et al., 2018), and the MineRL Navigate dataset (Guss et al., 2019). We evaluate open-loop video predictions on these datasets using four quantitative metrics: Structural Similarity index (SSIM; higher is better), Peak Signal-to-Noise Ratio (PSNR; higher is better), Learned Perceptual Image Patch Similarity (LPIPS; lower is better) (Zhang et al., 2018), and Frechet Video Distance (FVD; lower is better) (Unterthiner et al., 2018). In Section 4.5, we investigate how the amount of information stored at different levels of a temporal hierarchy adapts to changes in sequence speed. In Section 4.6, we visualize the information stored at different levels by resetting individual levels of the hierarchy.

We trained all our models using sequences of length 100. We used convolutional frame encoders and decoders, with architectures very similar to the DCGAN (Radford et al., 2016) discriminator and generator, respectively. Our implementations made use of TensorFlow Probability (Dillon et al., 2017) and CuDNN, and used the Adam optimizer (Kingma & Ba, 2014) for training. The training time for a 3-level TALD model with temporal abstraction 6 amounted to around 24 hours for 100 epochs on a single NVIDIA TITAN Xp GPU. Refer to Appendix C for hyperparameters and experimental setup.

## 4.1 MOVING MNIST DATASET

The Moving MNIST dataset consists of two digits moving in a square with velocities sampled in the range of 2 to 6 pixels per frame. We trained different versions of TALD with 3 levels in the hierarchy and temporal abstraction factors 1, 2, 4, 6 and 8, all of which use the same number of model parameters. We compare samples of long-horizon open-loop video predictions of 900 frames with RSSM and SVG-LP in Figure 4. All samples were conditioned using posterior beliefs inferred after observing 36 context frames. Please refer to Appendix C for more details and discussion.

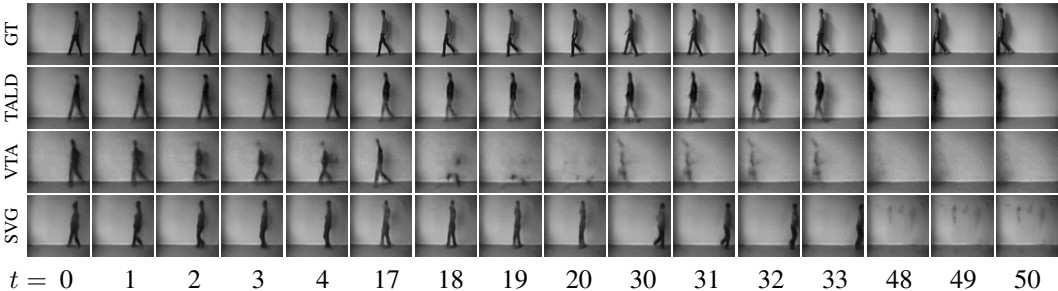

Figure 5: Open-loop video prediction for the KTH Action dataset. While TALD predicts accurately for 50 time frames, we observe jumpy transitions in VTA, where in this example the person disappears after the 17th frame. SVG predicts accurately for 18 frames, but starts to forget the task thereafter, as the person in the video starts to move in the opposite direction.

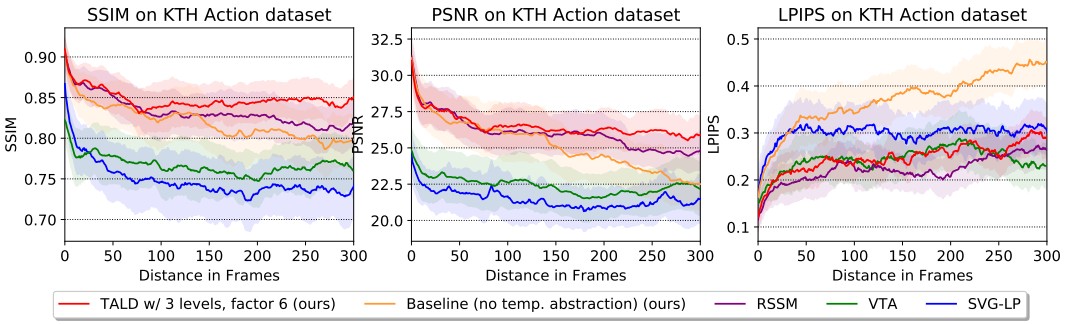

Figure 6: Quantitative comparison between our temporally abstract latent dynamics model (TALD) and baselines over open-loop video prediction of 300 frames for the KTH Action dataset.

We observe that SVG typically forgets object identity within 50 timesteps, while TALD with abstraction factor 6 maintains digit identity over 900 timesteps. RSSM clearly outperforms SVG, however starts to forget object identities after 250 time frames. TALD with abstraction factor 1 (i.e. no temporal abstraction) also starts to forget object identity after around 600 frames. With regards to the object positions, TALD with abstraction factor 6 predicts accurate digit positions until around 90 steps, and predicts a plausible sequence thereafter. RSSM and TALD without temporal abstraction predict the correct location of digits for at least as long as TALD with temporal abstraction. However, SVG starts to lose track of positions much sooner. We also note that our predictions are a bit blurry compared to those generated by SVG. Please refer to Appendix D for more experimental results.

We report the KL divergence value per level (summed over active time steps) for our TALD models in Table 1. Each value was obtained after training over sequences of length 100, for 200 epochs. The 2 and 3-level models were trained with a temporal abstraction factor of 6, and the 4-level model with a factor of 4 (to fit into memory). Figure 1 compares the Structural Similarity index (SSIM) for different versions of TALD with RSSM and SVG-LP. We note that SSIM decreases at a lower rate for models with higher temporal abstractions. As a baseline, we compute SSIM between ground truths and random sequences from the training set. It is interesting to note that quality of video predictions from TALD stay better than random for a 6 times longer duration than SVG.

## 4.2 KTH ACTION DATASET

We trained a 3-level TALD model with temporal abstraction factor 6 for the KTH Action dataset. In Figure 6, we report the SSIM, PSNR, and LPIPS, of TALD compared to SVG-LP, RSSM, and VTA. We also illustrate open-loop video predictions in Figure 5, conditioned using 36 context frames. While TALD predicts plausible frames for 50 timesteps, we observe jumpy transitions with VTA, probably because of breaks in the transition chain at the lower level. We also observe that SVG predicts accurately for 18 frames, while switching to a different task thereafter. We also note that SVG uses the DCGAN architecture for MNIST and the much larger VGG for KTH, whereas TALD works well even with the smaller DCGAN encoder/decoder. Refer to Appendix D for more results.

Figure 8: KL divergence at each level of the TALD model (3 levels, temporal abstraction factor 6), trained on slower/faster versions of Moving MNIST. Observe that the KL term at higher levels decreases with an increase in the speed of the digits, suggesting less global information being pushed up in the hierarchy.

## 4.3 GQN 3D MAZES DATASET

We trained a 2-level TALD model with temporal abstraction factor 6, and compared it with RSSM and VTA, on the GQN mazes dataset. Figure 7 shows open-loop video prediction samples, conditioned using 36 context frames. We observe that while our model can maintain global information of wall and floor colors for 200 frames, RSSM starts to forget the same after ∼50 frames. Even though the open-loop predictions from TALD differ from ground truth in terms of camera viewpoints, the model does not forget the wall and floor patterns. We

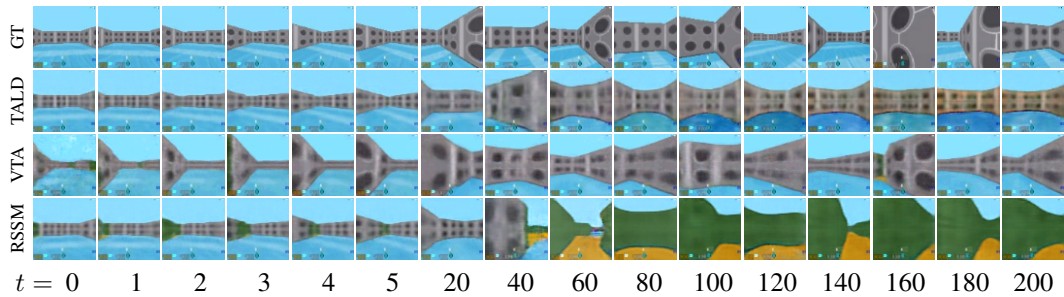

Figure 7: Long horizon open-loop video prediction for the GQN mazes dataset (Eslami et al., 2018) using our 2-level TALD model with temporal abstraction factor of 6. RSSM does not use temporal abstraction and thus starts to forget wall and floor patterns after 40 frames, while TALD maintains this global information over 200 frames.

compare quantitative metrics on this dataset in Figure 12d. Please refer to Appendix D for detailed experimental results.

### 4.4 MineRL Navigate dataset

We trained a 3-level TALD model with temporal abstraction factor 6, and compared it with RSSM, SVG-LP and VTA, on a Minecraft dataset (MineRL Navigate) (Guss et al., 2019). This dataset features videos in a variety of world environments with complex moving backgrounds. We show long-horizon open-loop video predictions upto 420 frames (conditioned on 36 context frames) on this dataset in Figure 2. We also compare quantitative metrics on this dataset in Figure 12c in Appendix D. Please refer to Figure 15 in Appendix D for more open-loop video prediction examples.

### 4.5 Adapting to Sequence Speed

In order to understand how our model adapts to changing temporal correlations in the dataset, we trained our model with slower and faster versions of moving MNIST, with speeds varied by factors of 3. For this experiment, our model consisted of 3 levels in the hierarchy, with each level temporally abstracting the level below by a factor of 6.

Figure 8 shows the KL divergence summed across the active timesteps at every level in the hierarchy. We observe that there is a correlation between the KL divergence at every level and the speed at which the digits move. There is more information at level 1 when the digits move faster, and consequently lesser information at the levels above it. Also, even though the KL divergence at level 3 is small, it still follows the same trend as the other two levels. It is also important to note that the KL divergence between the prior and the posterior is only an upper bound on the information stored by the encoder in a posterior belief state.

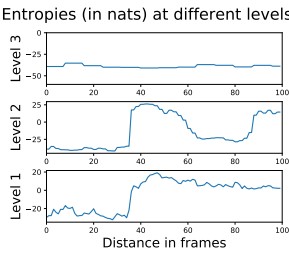

(a) GQN mazes

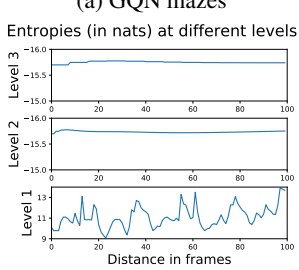

(b) Moving MNIST

Figure 10: Entropy of the prior during open-loop video generation at different levels of TALD (3 levels, temporal abstraction 2). (a) With GQN mazes, constant entropy at level 3 (stored wall and floor patterns). (b) With Moving MNIST, constant entropy at level 2 (stored digit identities), and level 3 (suffered posterior collapse).

### 4.6 Resetting Individual Levels

We visualize the information stored at a certain level by replacing the posterior belief at one level with the prior belief, i.e. all but one level receive observations (Zhao et al., 2017). Conditioned on those posterior beliefs, we sample open-loop video predictions using the trained prior model, which should show variations in the attributes learned at that level. We expect our model to store global information high up in the hierarchy, allowing the model to perform fewer transitions over that information, making it easier to pay less cost in the form of KL divergence during training.

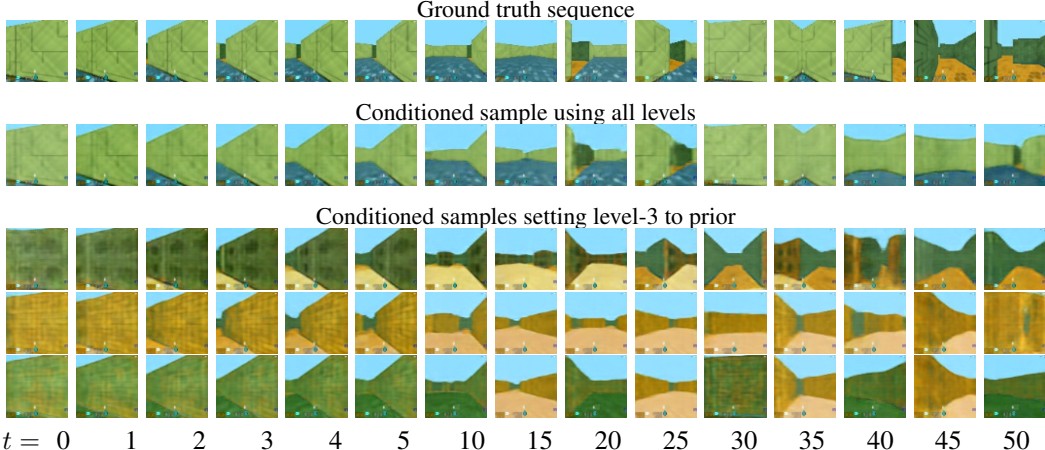

Figure 9: Visualizing the information stored at the higher level of our 3-level model with temporal abstraction factor 2, using the GQN mazes dataset. We computed a posterior belief at each level using 8 observation frames, and set one of the levels to the prior (by not feeding it with observations), which were then used to condition open-loop predictions. Changing wall textures show that they are stored at the highest level.

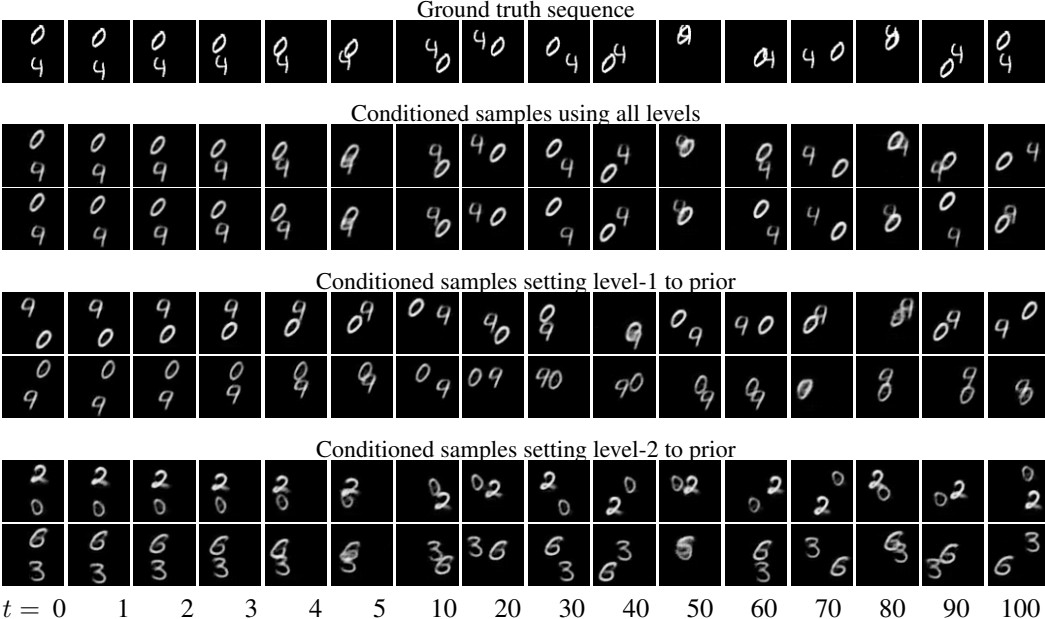

Figure 11: Visualizing the information stored at different levels of TALD with 3-levels and temporal abstraction factor 6, for Moving MNIST. We computed a posterior belief at each level using 36 observation frames, while setting one of the levels to the prior (by not feeding it with observations), which were then used to condition open-loop predictions.

Figure 9 shows video predictions with different levels reset to the prior for the GQN mazes dataset. With a 3-level model and temporal abstraction factor 2, we observe that when level 3 was not fed with observations, the conditioned open-loop predictions started with the correct viewpoint, but differed w.r.t. wall and floor colors. This suggests that those characteristics were stored in the higher level of the hierarchy.

Figure 11 shows video predictions with different levels reset to the prior for the Moving MNIST dataset. The 3-level model with temporal abstraction factor 6 obtained a separation of information in the bottom two levels, while the posterior at the third level nearly collapsed to the prior. When level 1 was not fed with observations, we observe that the conditioned open-loop predictions maintained the same digit identity, but showed variations w.r.t. digit positions. On the other hand, when level 2

was not fed with observations, the samples maintained the same digit positions but produced digits with different identities in every sample. This suggests that lower level stored digit positions, high frequency details which changed frequently in time, while the level above it stored the digit identities, i.e. long-term information. We also observed that, when resetting level 2 to the prior, the digits start to differ in position sooner ($\sim$60 frames) than when all levels receive observations ($\sim$80 frames). This suggests that this level does have some information about digit positions, and that there is still mixing of information between different levels of the hierarchy.

**Predictive entropy**    To corroborate our understanding of the latent representations, we observe the entropy of the prior distribution as it varies over time during open-loop video generation in Figure 10. With GQN mazes, the entropy at the top level remained relatively constant as the model remained certain about the high-level details. With Moving MNIST, the top two levels showed a relatively constant entropy, with level 2 storing the digit identities and level 3 suffering from posterior collapse. Please refer to Appendix D.6 for a more detailed analysis.

## 5    DISCUSSION

In this work, we presented a hierarchical latent dynamics model with temporal abstraction (TALD), where each level in the hierarchy temporally abstracted the level below by an adjustable factor.

- We evaluated long-horizon open-loop predictions using our model, and observed that TALD was able to predict far into the future while accurately maintaining global information.

- We also observed that the amount of information at the higher levels decreased as the speed of the sequence was increased.

- We analyzed the separation of information at different levels of the hierarchy, by generating open-loop video predictions with different levels reset to the prior. With Moving MNIST, the bottom level in the hierarchy stored high frequency details (digit positions) and the level above stored more global information (digit identities). With the GQN mazes dataset, TALD stored wall and floor patterns at the top level in the hierarchy.

Temporally abstract models are an intuitive approach to obtaining high-level representations of complex datasets and environments. We hope that our work can refuel interest in temporally abstract latent dynamics models and motivate the development of effective deep learning systems for high-dimensional data more generally.

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

## A    BACKGROUND

We build our work upon the recurrent state-space model (RSSM; Hafner et al., 2019) that has been shown to successfully learn environment dynamics from raw pixels in reinforcement learning. This model acts as an important baseline for our evaluation. RSSM explains the video sequence $x_{1:T}$ using a latent sequence of compact Markovian states $s_{1:T}$. Importantly, the model is autoregressive in latent space but not in image space, allowing us to predict into the future without generating images along the way,

$$p(x_{1:T}, s_{1:T}) \doteq \prod_{t=1}^{T} p(x_t \mid s_t) p(s_t \mid s_{t-1}). \tag{6}$$

Given a training sequence, RSSM first individually embeds the frames using a CNN. A recurrent network with deterministic and stochastic components then summarizes the image embeddings. Hafner et al. (2019) argued that the stochastic component helps with modeling multiple futures, while the deterministic state helps remember information over many timesteps so that it is not erased by noise. Finally, the recurrent states are decoded using a transposed CNN to provide a training signal.

$$
\begin{aligned}
\text{Encoder:} & \quad e_t = \text{enc}(x_t) \\
\text{Posterior transition } q_t\text{:} & \quad q(s_t \mid s_{t-1}, e_t) \\
\text{Prior transition } p_t\text{:} & \quad p(s_t \mid s_{t-1}) \\
\text{Decoder:} & \quad p(x_t \mid s_t).
\end{aligned}
\tag{7}
$$

The posterior and prior transition models share the same recurrent model. The difference is that the posterior incorporates images, while the prior tries to predict ahead without knowing the corresponding images. This lets us to predict ahead purely in latent space at inference time.

As typical with deep latent variable models, we cannot compute the likelihood of the training data under the model in closed form. Instead, we use the evidence lower bound (ELBO) as training objective. The ELBO encourages to accurately reconstruct each image from its corresponding latent state, while regularizing the latent state distributions to stay close to the prior dynamics,

$$\max_{q,p} \sum_{t=1}^{T} \mathbb{E}_{q_t}[\ln p(x_t \mid s_t)] - \sum_{t=1}^{T} \text{KL}[q_t \parallel p_t]. \tag{8}$$

The KL regularizer limits the amount of information that the posterior transition incorporates into the latent state at each time step, thus encouraging the model to mostly rely on information from past time steps and only extract information from each image that cannot be predicted from the preceding images already. All components jointly optimize Equation 8 using stochastic backpropagation with reparameterized sampling (Kingma & Welling, 2013; Rezende et al., 2014).

## B    MODEL ARCHITECTURES

We use convolutional frame encoders and decoders, with architectures very similar to the DCGAN (Radford et al., 2016) discriminator and generator, respectively. To obtain the input embeddings $e_t^l$ at a particular level, $k^{l-1}$ input embeddings are pre-processed using a feed-forward network and then added up to obtain a single embedding. We also want to emphasize that we do not use any skip connections between the encoder and decoder bypassing the latent states as we believe this would motivate the model to make better use of the temporal hierarchy.

## C    HYPER PARAMETERS AND EXPERIMENTAL SETUP

We kept the output dimensionality of the encoder at each level of TALD as $|e_t^l| = 1024$, that of the stochastic states as $|p_t^l| = |q_t^l| = 20$, and that of the deterministic states as $|h_t^l| = 200$. All hidden layers inside the cell, both for prior and posterior transition, were set to 200. We trained our models using the Adam optimizer (Kingma & Ba, 2014), with a learning rate of $5 \times 10^{-4}$ and $\epsilon = 10^{-4}$ for the Moving MNIST dataset. With the KTH Action and GQN mazes datasets, we lowered the learning rate to $3 \times 10^{-4}$. For all three datasets, batch size was set to be 100 sequences, each sequence being of length 100.

The decision of using 36 context frames for conditioning all open-loop video predictions was taken considering the minimum number of observation frames required to transition at least once in the highest level of the hierarchy. With 3 levels in the hierarchy and a temporal abstraction factor of 6, each latent state at the highest level corresponds to 36 images in the sequence, and thus its encoder network expects 36 images as input.

# D  ADDITIONAL EXPERIMENTS

## D.1  QUANTITATIVE METRICS FOR LONG-HORIZON VIDEO PREDICTION

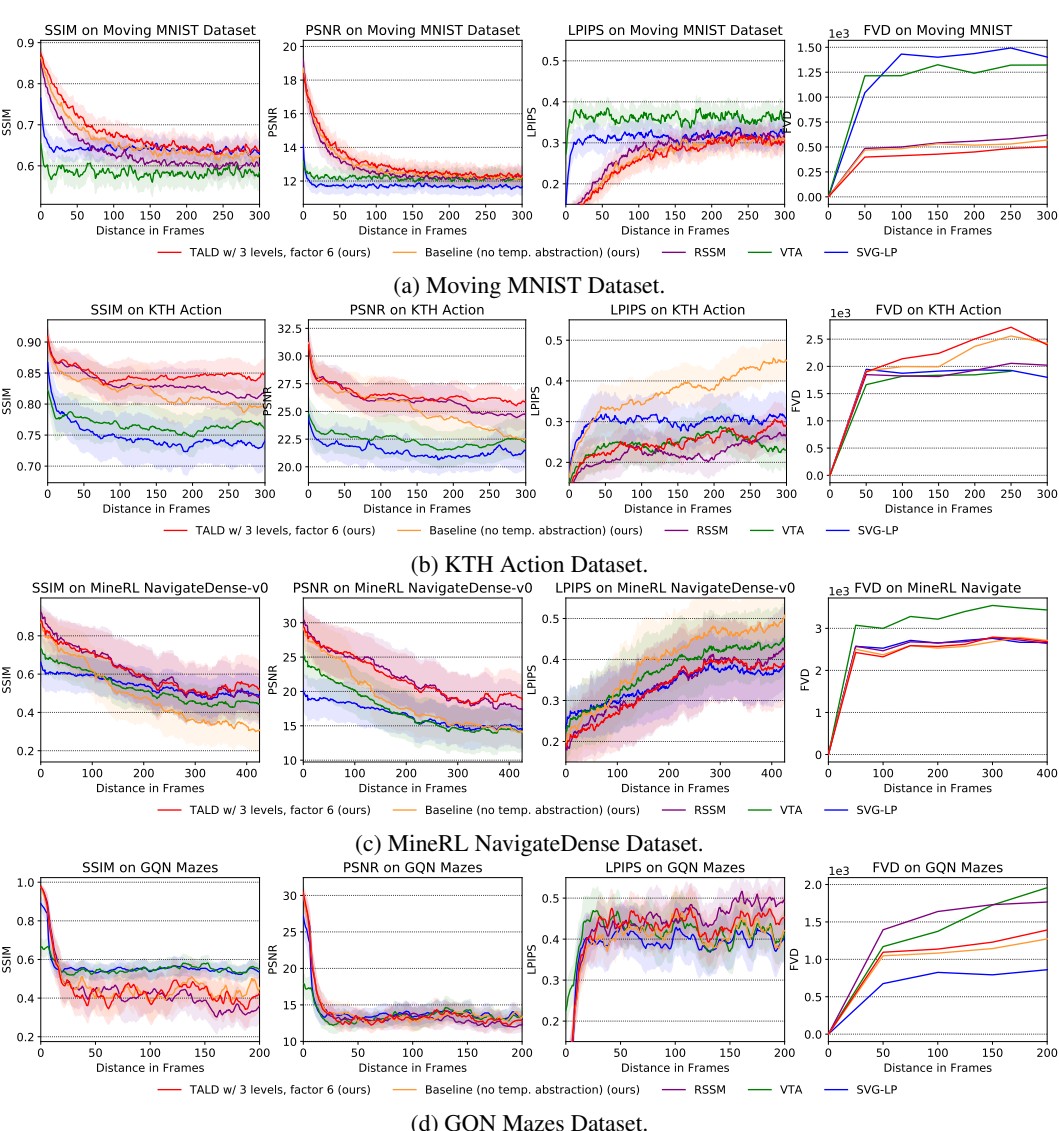

Figure 12: Quantitative comparison between temporally abstract latent dynamics model (TALD) and other video prediction models, for long-horizon open-loop video prediction. Higher values are preferred for SSIM and PSNR, whereas lower values are preferred for LPIPS and FVD.

## D.2 LONG-HORIZON PREDICTION OF MOVING MNIST

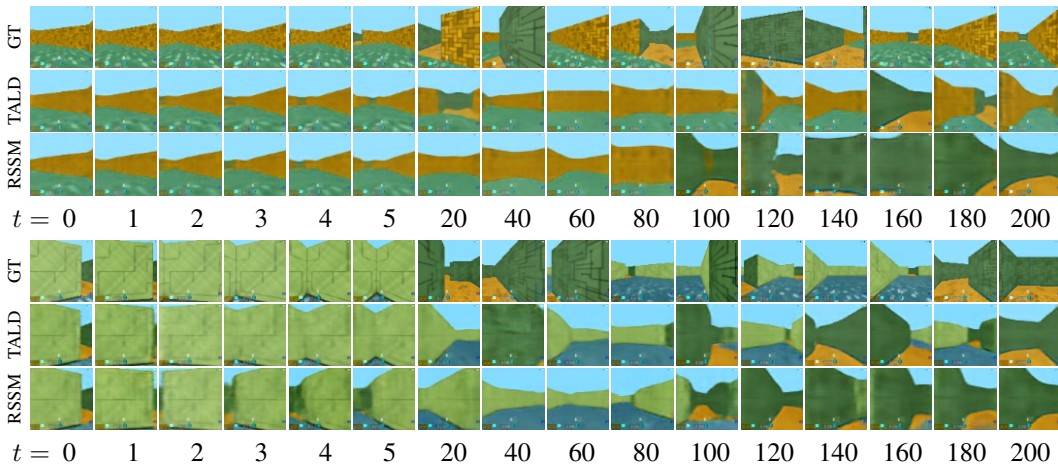

Figure 13: Long-horizon open-loop prediction on Moving MNIST. (L for levels, F for abstraction factor.) The top row shows the ground truth sequence. The next two rows illustrate samples generated from our TALD model: 3 levels with temporal abstraction factor of 6 (3L-F6), and 3 levels with temporal abstraction factor of 1 (3L-F1) (essentially a deeper model with no temporal abstractions). Following that we have samples from the baselines of RSSM and SVG-LP. It is interesting to note here that TALD with an abstraction factor 6 is able to maintain accurate long-term correlation in the form of object identities for 900 frames into the future.

## D.3 LONG-HORIZON PREDICTION OF 3D MAZES

Figure 14: Long horizon open-loop video prediction for the GQN mazes dataset (Eslami et al., 2018) using our 2-level TALD model with a temporal abstraction factor of 6. In both the examples, the mazes switch between two types of wall patterns as the camera moves around the maze. We observe that while TALD could remember the alternate wall patterns for 200 timesteps, RSSM could not predict any frame with the previously generated wall patterns. This suggests that TALD was able to maintain long-term information for at least 200 frames.

## D.4 LONG-HORIZON PREDICTION OF MINERL NAVIGATE

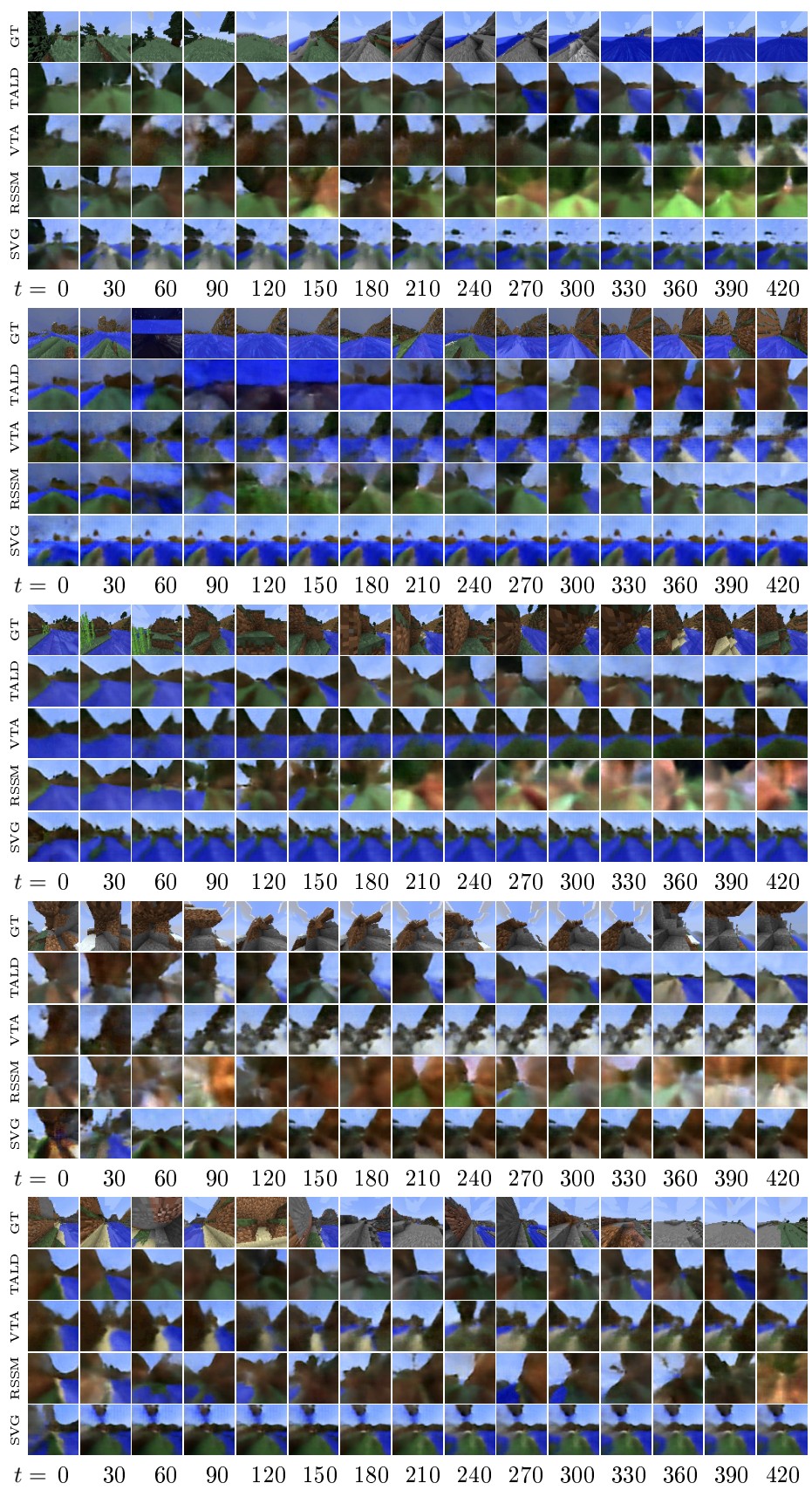

Figure 15: Long-horizon open-loop video prediction for the MineRL Navigate dataset (Guss et al., 2019), using our 3-level TALD model with a temporal abstraction factor of 6, along with some established baselines.

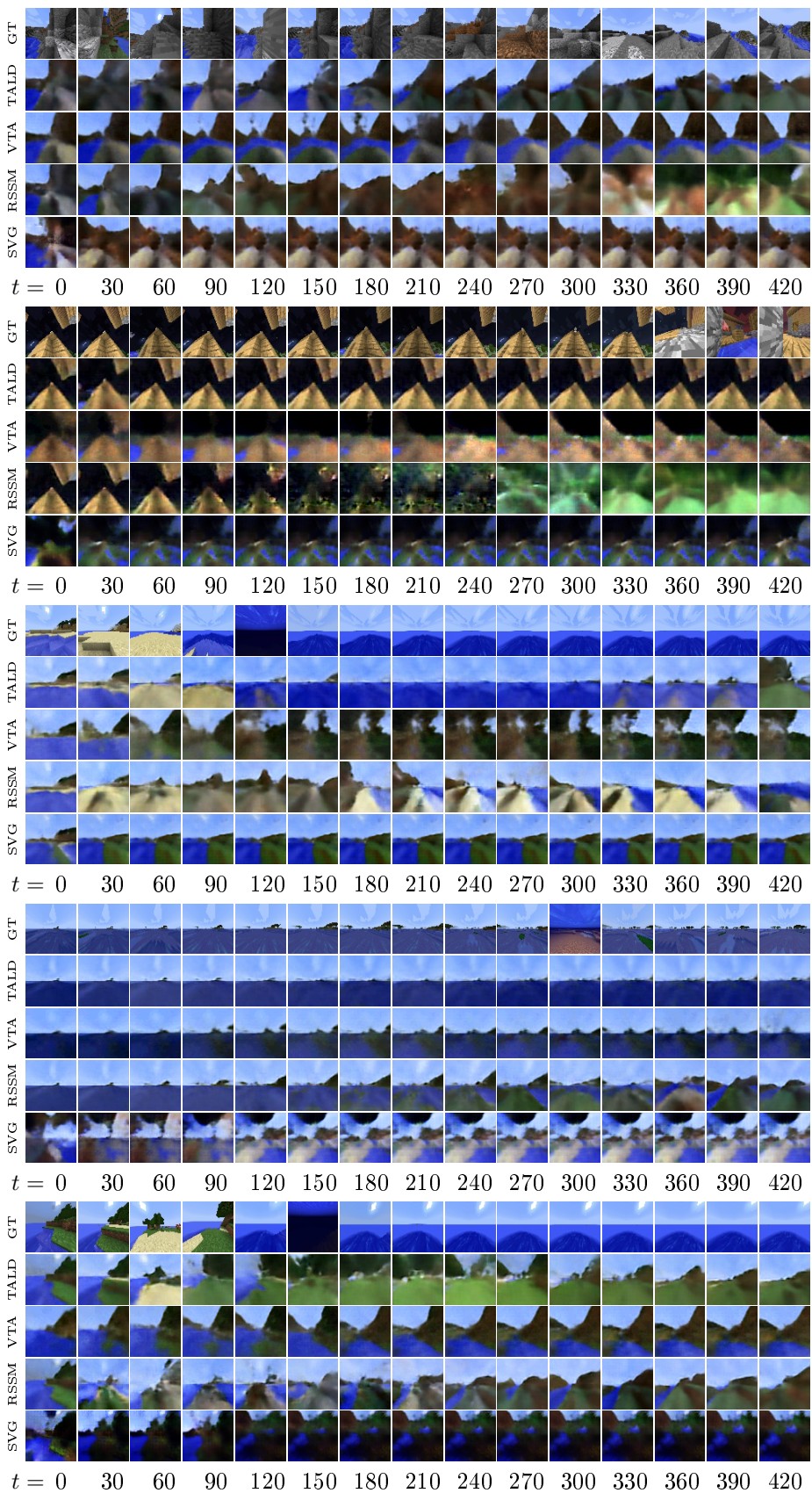

Figure 15: (contd.) Long-horizon open-loop video prediction for the MineRL Navigate dataset (Guss et al., 2019), using our 3-level TALD model with a temporal abstraction factor of 6, along with some established baselines.

## D.5 Understanding the Latent Representations - 3D Mazes

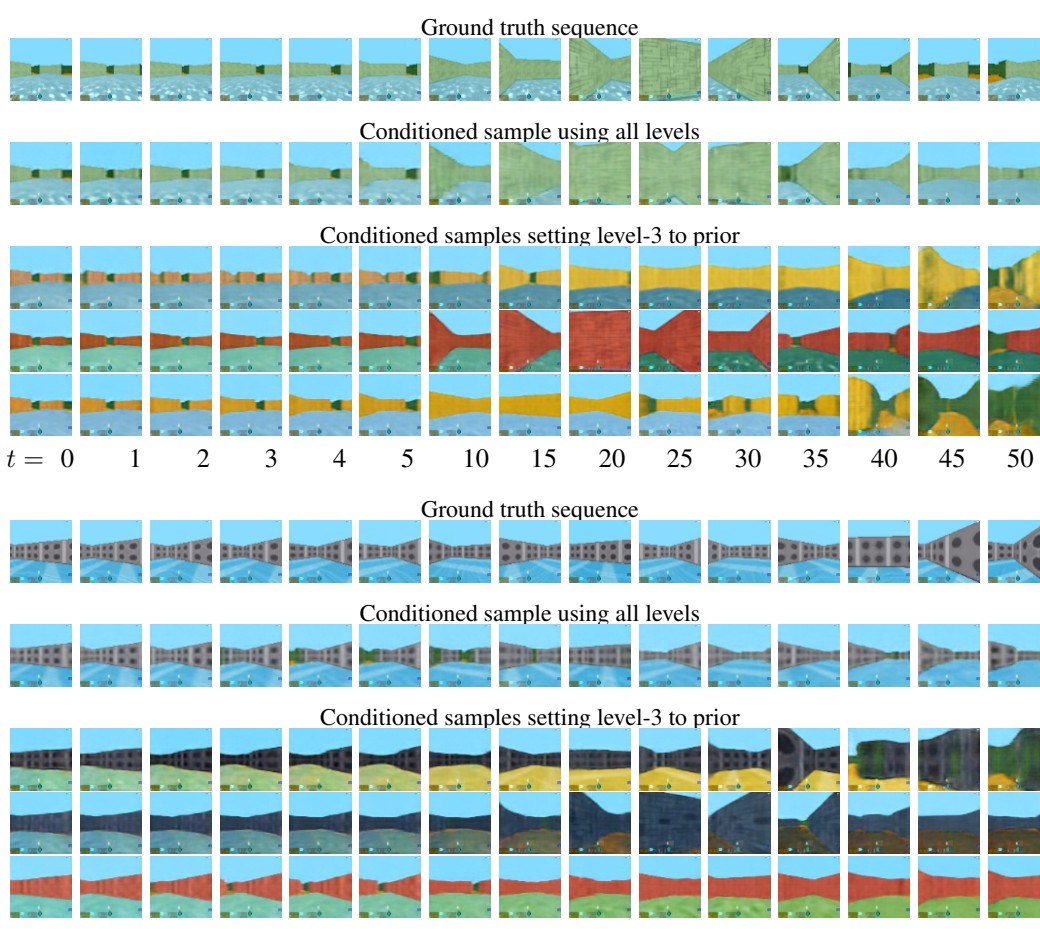

Figure 16: Visualizing the information stored at the higher level of our 3-level model with temporal abstraction factor 2, using the GQN mazes dataset. We computed a posterior belief at each level using 8 observation frames, and set one of the levels to the prior (by not feeding it with observations), which were then used to condition open-loop predictions. Changing wall textures show that they are stored at the highest level.

## D.6 Analysing the Entropy Curves

Figure 10b shows the entropy of the prior distribution at all levels of the 3-level TALD model with temporal abstraction factor 2, as it varies over time, with the Moving MNIST dataset. We observe that the entropy at the lowest level varies between 9 to 13 nats, whereas for the higher levels in the hierarchy, the entropy remained relatively constant. In Figure 11 we presented some visualizations that supported the claim that the higher level in the hierarchy stores global information (digit identities) which, in this case, does not change over time, and hence the model should remain rather certain about as time progresses. Figure 10b corroborates that claim showing that the model's uncertainty at higher levels does not dither significantly over time as compared to the lowest level in the hierarchy.

Figure 10a shows the entropy of the prior distribution at all levels of the 3-level TALD model with temporal abstraction factor 2, as it varies over time, with the GQN mazes dataset. While the bottom two levels showed a similar variance in entropy with time, the entropy of the prior distribution at the top level did not change significantly. We showed in Figure 9 how the top level in the hierarchy stores wall and floor colors, which the model should remain certain about over time. In support of this claim, here we also see that the entropy of the prior distribution at the topmost level does not change significantly over time.

