# OpenReview forum: "Video Prediction with Variational Temporal Hierarchies"
_ICLR.cc/2021/Conference — Reject_

### Official Review · AnonReviewer4 · 2020-10-27
**An interesting paper with a need for stronger experiments**

**Rating:** 5
**Confidence:** 4

**Review:**

The authors present build on the recurrent state space model by Haffner et al. (2019) to include a hierarchy of latent spaces. The idea is that the lower levels model short-term changes while the higher levels of the hierarchy model longer temporal dependencies.

Pros:
1. The paper is largely well-written and easy to follow.
2. The experiments do show that the proposed method is able to capture the long-term trends better than the chosen baselines for the chosen datasets.
3. Experiments showing different amounts of information present at different levels of the latent space are interesting and show that desired hierarchical structure in the latent space is achieved.

Cons:
1. In light of the paper "Improved conditional VRNNs for video prediction" by Castrejon et al., I do not believe there is enough novelty in this paper in terms of the latent space architecture, as this paper also applies a hierarchical latent space model for video prediction. The main contributions are in the analysis of the hierarchical architectures using information-theoretic measures for three datasets, which I believe are not sufficient for accepting this paper. The authors should clarify the main differences in the proposed method and the paper by Castrejon et al.

2. Related to the above point, the datasets for the experiments should be stronger as this paper's novelty is mainly empirical. The KTH action dataset is the closest to a real-world dataset, but is still simplistic with just one moving object and a fixed background. The authors should consider experimenting with more challenging datasets. This is necessary to understand the limitations of the proposed method.

UPDATE AFTER AUTHOR RESPONSE:
I appreciate the authors' response and for clarifying the contributions. However I still feel that the experiments and datasets are too simple and not realistic. I am increasing my rating to reflect this.

---

> ### Author Response · Authors · 2020-11-15
> **Authors' Response: Added experimental results**
>
> We highly appreciate your time spent reviewing our paper and providing insightful comments!
>
> We address your two concerns with our submission below. We believe that this will resolve the two concerns you mentioned, but if not, please let us know and we would be happy to engage in discussion and perform additional experiments.
>
> We have updated our paper to include Minecraft as an additional challenging long-range video prediction dataset. The data was collected by Guss et al. (2019) and contains sequences of human players who navigate to a goal location in a 3D procedurally generated world (traversing forests, mountains, villages, oceans, etc). We include 150-step video predictions in Figure 14 in the appendix. We find that TALD and RSSM generate a plausible sequence for 150 frames into the future, with TALD doing slightly better on three quantitative metrics. In contrast, SVG-LP learns to copy the initial frames indefinitely and does not predict any new events in the future, while VTA generates implausible events over longer horizons.
>
> Moreover, we have included additional quantitative metrics and now report SSIM, PSNR, LPIPS, and FVD for all models and datasets in Figure 11 in Appendix D.
>
>
> **Comment:** The authors should clarify the main differences in the proposed method and the paper by Castrejon et al.
>
> Our work focuses on temporal abstraction. In contrast, the hierarchical VRNN proposed by Castrejon et al. only employs spatial abstraction.
>
> In our experiments, we compare to one such model that leverages hierarchy only for spatial but not temporal abstraction (3-level TALD model with temporal temporal abstraction factor “1”, which has the same number of parameters as any other 3-level TALD with more temporal abstraction). The experiments on all four datasets clearly highlight the importance of temporal abstraction for accurate long-term predictions.
>
> Another difference is that they build upon VRNN which feeds generated images back into the latents, while we build upon RSSM which predicts forward purely in latent space. This enables fast predictions with small memory footprint for downstream applications, such as Hafner et al. (2019).
>
> **Comment:** The authors should consider experimenting with more challenging datasets.
>
> Thank you for this suggestion. To address this feedback, we have selected the Minecraft Dataset (Guss et al., 2019) as a fourth and substantially more challenging dataset. We compare TALD, RSSM, SVG-LP, and VTA on this dataset.
>
> In summary, we find that TALD was able to predict plausible sequences longer than some baselines. We report all quantitative metrics on this dataset in Figure 11 of Appendix D. Please refer to Section 4.4 and Appendix D in the updated PDF for complete experimental details and results.

---

### Official Review · AnonReviewer3 · 2020-10-28
**A sequential hierarchical VAE for long-term video prediction. Quantitative evaluation could be improved.**

**Rating:** 5
**Confidence:** 4

**Review:**

This paper tackles the problem of long term video prediction. It proposes a novel sequential hierarchical VAE-based model called Temporal Abstract Latent Dynamics (TALD), that can keep long term consistency by having different levels of the hierarchy update at different frequencies.
Empirically, the authors validate that the higher levels of the hierarchy do indeed encode for slower changing features in the videos. They also show that using temporal abstraction allows TALD to produce better predictions of long sequences (hundreds of frames) and outperform prior models in that setting.

################################################


Strong points:

- The paper is clear and easy to follow.

- It deals with an important topic, as long-term consistency is a challenge in many settings for sequential deep learning models.

- The idea behind the model is simple and appealing.

- Thorough experimental results validate that the model behaves as expected.


Concerns:

- Mainly, evaluation of performance seems inadequate:

All of the considered models (including the proposed one) are of stochastic nature. They are trained to provide plausible sequence, and not retrieve the exact ground-truth sequence. Yet, the proposed evaluation metric ignores the stochastic aspect, assuming the models should be able to correctly retrieve the ground-truth sequence given the initial frames.
Those metrics might be ill-suited, as small positional shift due to the stochasticity can propagate through time and lead to huge drops in performances even if the digits are correctly conserved the entire sequence and the trajectories mostly correct. This actually shows up when comparing Figure 1 (where all models have scores comparable to random frames at the end)  and Figure 4 that shows that the digits mostly remain unchanged.
Why weren't stochastic variants of PSNR, LPIPS and SSIM (as used in the original publication for SVG-LP), or Frechet Video Distance (maybe on subsampled frames of the sequences) considered instead?

Also, the long-term consistency is mostly evaluated by human observation on samples. Maybe, a digit detector (and a wall/ground texture detector) could be used to provide a quantitative evaluation?

################################################

Score motivation:

The paper tackle an important problem with a well-motivated novel model. However, I have concerns about the quantitative evaluation methods.

################################################

Minor remarks:

- How are the initial states h_0 computed? This information might be needed for reproducibility.


The authors might also want to acknowledge (or compare to) some recent works in video generation:

- Stochastic Adversarial Video Prediction
Alex X. Lee, Richard Zhang, Frederik Ebert, Pieter Abbeel, Chelsea Finn, Sergey Levine

- VideoFlow: A Conditional Flow-Based Model for Stochastic Video Generation
Manoj Kumar, Mohammad Babaeizadeh, Dumitru Erhan, Chelsea Finn, Sergey Levine, Laurent Dinh, Durk Kingma

- Stochastic Latent Residual Video Prediction
Jean-Yves Franceschi, Edouard Delasalles, Mickael Chen, Sylvain Lamprier, Patrick Gallinari

- Scaling Autoregressive Video Models
Dirk Weissenborn, Oscar Täckström, Jakob Uszkoreit

---

> ### Author Response · Authors · 2020-11-15
> **Authors' Response: Added experimental results**
>
> We highly appreciate your time spent reviewing our paper and providing insightful comments!
>
> We address your concerns with our submission below. We believe that this will resolve the concerns you mentioned, but if not, please let us know and we would be happy to engage in discussion and perform additional experiments.
>
> We have updated our paper to include Minecraft as an additional challenging long-range video prediction dataset. The data was collected by Guss et al. (2019) and contains sequences of human players who navigate to a goal location in a 3D procedurally generated world (traversing forests, mountains, villages, oceans, etc). We include 150-step video predictions in Figure 14 in the appendix. We find that TALD and RSSM generate a plausible sequence for 150 frames into the future, with TALD doing slightly better on three quantitative metrics. In contrast, SVG-LP learns to copy the initial frames indefinitely and does not predict any new events in the future, while VTA generates implausible events over longer horizons.
>
> Moreover, we have included additional quantitative metrics and now report SSIM, PSNR, LPIPS, and FVD for all models and datasets in Figure 11 in Appendix D.
>
>
> **Comment:** Why weren't stochastic variants of PSNR, LPIPS and SSIM (as used in the original publication for SVG-LP), or Frechet Video Distance (maybe on subsampled frames of the sequences) considered instead?
>
> To address your concern, we evaluated our model and baselines on the Frechet Video Distance (FVD) (on subsequences of length 50), along with the existing SSIM, PSNR, and LPIPS metrics, for all datasets. We provide detailed experimental results on all datasets in Appendix D of the paper.
>
> **Comment:** Also, the long-term consistency is mostly evaluated by human observation on samples. Maybe, a digit detector (and a wall/ground texture detector) could be used to provide a quantitative evaluation?
>
> We illustrate quantitative metrics (SSIM, PSNR, LPIPS, and FVD) over long-horizons to evaluate the long-term video prediction capabilities of these models. We agree that we measure high-level semantics in video through observation. However, at this point, we leave for future work to develop a quantitative metric to measure the quality of semantics in video as they vary over time for these datasets.
>
> **Comment:** The paper tackle an important problem with a well-motivated novel model. However, I have concerns about the quantitative evaluation methods.
>
> We have updated our experimental results with quantitative evaluations on all existing datasets, along with a substantially more complex Minecraft dataset (Guss et al., 2019). Please find a short description of our experiments on this dataset in Section 4.4, and quantitative metrics in Appendix D. We would be happy to address any comments and perform additional experiments as needed.
>
> **Comment:** How are the initial states h_0 computed? This information might be needed for reproducibility.
>
> The initial deterministic and stochastic states, h_0 and z_0, for each level in TALD, were inferred using all available observation frames. The inference of a state at every level was conditioned on the observations within the scope of that level at that time-frame, and the inferred state at the level above. The inferred states at the last observation frame were used as initial states for subsequent open-loop generation. We will eventually open-source our code to make this implementation more clear.
>
> **Comment:** The authors might also want to acknowledge (or compare to) some recent works in video generation...
>
> Thank you for pointing out these recent works. As per your suggestion, we have updated our Related Works section with a discussion on these papers.

---

### Official Review · AnonReviewer2 · 2020-10-29
**Well-experimented paper with limited novelty**

**Rating:** 4
**Confidence:** 5

**Review:**

This paper presents a recurrent network for long-term video prediction, which learns hierarchical latent states in space-time. Although it shows strong experimental results, I have the following concerns about the novelty of the proposed model and the completeness of the experiments:
1. To me, the proposed TALD architecture can be seen as a combination of two previous models, RSSM and VTA, with small extensions. While I understand that they have many differences in technical details, such as the number of layers and the dependencies between the latent states, the key insight behind these approaches is similar.
2. Hierarchical latent variables have long been used for video prediction, as in Wichers et al. (2018) and Xu et al. (2018). I think the authors should make more comparisons with these similar works both in terms of methods and experimental results.
[Xu et al., 2018] PredCNN: Predictive Learning with Cascade Convolutions
3. For both Moving MNIST and KTH datasets, the proposed model tasks as input 36 context frames, which is not a common practice of most previous models, including those from Wichers et al. (2018) and Denton et al. (2018). Given that the temporal dynamics is relatively simple in these datasets, it may not be necessary to use so much context information. What would happen if we reduce the number of input frames?
4. The authors put the descriptions of Figure 1 in Section 4.1 and the descriptions of Figure 2 in Section 4.3, which caused me some reading difficulties at the very beginning of the paper.

---

> ### Author Response · Authors · 2020-11-15
> **Authors' Response**
>
> We highly appreciate your time spent reviewing our paper and providing insightful comments!
>
> We have updated our paper to include Minecraft as an additional challenging long-range video prediction dataset. The data was collected by Guss et al. (2019) and contains sequences of human players who navigate to a goal location in a 3D procedurally generated world (traversing forests, mountains, villages, oceans, etc). We include 150-step video predictions in Figure 14 in the appendix. We find that TALD and RSSM generate a plausible sequence for 150 frames into the future, with TALD doing slightly better on three quantitative metrics. In contrast, SVG-LP learns to copy the initial frames indefinitely and does not predict any new events in the future, while VTA generates implausible events over longer horizons.
>
> Moreover, we have included additional quantitative metrics and now report SSIM, PSNR, LPIPS, and FVD for all models and datasets in Figure 11 in Appendix D.
>
>
> **Comment:** To me, the proposed TALD architecture can be seen as a combination of two previous models, RSSM and VTA, with small extensions. While I understand that they have many differences in technical details, such as the number of layers and the dependencies between the latent states, the key insight behind these approaches is similar.
>
> Temporal abstraction in latent variable models has been studied since the 1990s (Towards Compositional Learning in Dynamic Networks (Schmidhuber, 1990)). The current status of research in video prediction makes us realise that what matters in practice are exactly the technical details. Moreover, we not just propose a TALD but also focus on building an understanding of this class of models, through our experiments using datasets of varying speeds, showing KL divergence at different levels, predictive entropy, and per-level generative distributions.
>
> **Comment:** Hierarchical latent variables have long been used for video prediction, as in Wichers et al. (2018) and Xu et al. (2018). I think the authors should make more comparisons with these similar works both in terms of methods and experimental results. [Xu et al., 2018] PredCNN: Predictive Learning with Cascade Convolutions
>
> Wichers et al. (2018) proposed a hierarchical latent structure beneficial towards long-horizon video prediction. Xu et al. (2018) proposed an entirely CNN-based architecture for modeling dependencies between sequential inputs. These works provide meaningful contributions towards large-scale video prediction, however their focus differs from that of this work. This work focuses on understanding the qualitative and quantitative effects of temporal abstraction on long-term video prediction, which is why we felt an empirical comparison with these papers was not necessary. We have updated our Related Works section to include discussions about these papers.
>
> **Comment:** What would happen if we reduce the number of input frames?
>
> This is a purely practical choice that lets us reuse the encoder network to initialize the latent states during evaluation time.
>
> With 3 levels in the hierarchy and a temporal abstraction factor of 6, each latent state at the highest level corresponds to 36 images in the sequence. Its encoder network thus expects 36 images as input. Alternatively, one could infer the initial latent states without the encoder using online inference to avoid this limitation. We updated the PDF in Appendix C to explain this decision.
>
> **Comment:** The authors put the descriptions of Figure 1 in Section 4.1 and the descriptions of Figure 2 in Section 4.3
>
> We have swapped the figure placement to increase readability in the updated PDF.

---

### Official Review · AnonReviewer1 · 2020-10-29
**A good paper, novel idea, OK experiments**

**Rating:** 6
**Confidence:** 4

**Review:**

This paper proposes a method called Temporal Abstract Latent Dynamics (TALD). TALD is built up on RSSM (Hafner et al. 2019) but with hierarchical dynamics. The experiments are conducted on moving MNIST, GQN 3D Mazes, and KTH. Results are qualitatively better than other methods in term of maintaining long-term consistent prediction. Quantitative comparison is reported only on KTH dataset (Figure 5). Written presentation is clear and easy to understand.

Pros:
- The idea of modeling hierarchical latent variables for long-term video prediction is novel and interesting.
- TALD qualitatively and quantitatively (on KTH) outperforms other methods, such as RSSM (direct baselines), VTA, SVG.
- The experiments of removing one level of hierarchical latent (by replacing the posterior) is particularly interesting which proved that TALD can abstract different semantic information at different level its hierarchical representation.

Cons:
- The comparisons on moving MNIST and GQN Mazes are only qualitatively (which may be biased to the selected sequences). The supplementary provides two more sequences (1 from moving MINIST and 1 from GQN Mazes) with qualitative results, but not quantitative results on these benchmarks. It should be better to report some sort of quantitative comparisons and evaluated over the full dataset instead (as done with KTH).

---

> ### Author Response · Authors · 2020-11-15
> **Authors' Response: Added experimental results**
>
> We highly appreciate your time spent reviewing our paper and providing insightful comments!
>
> We address your concerns with our submission below. We believe that this will resolve the concerns you mentioned, but if not, please let us know and we would be happy to engage in discussion and perform additional experiments.
>
> We have updated our paper to include Minecraft as an additional challenging long-range video prediction dataset. The data was collected by Guss et al. (2019) and contains sequences of human players who navigate to a goal location in a 3D procedurally generated world (traversing forests, mountains, villages, oceans, etc). We include 150-step video predictions in Figure 14 in the appendix. We find that TALD and RSSM generate a plausible sequence for 150 frames into the future, with TALD doing slightly better on three quantitative metrics. In contrast, SVG-LP learns to copy the initial frames indefinitely and does not predict any new events in the future, while VTA generates implausible events over longer horizons.
>
> Moreover, we have included additional quantitative metrics and now report SSIM, PSNR, LPIPS, and FVD for all models and datasets in Figure 11 in Appendix D.
>
>
> **Comment:** The supplementary provides two more sequences (1 from moving MINIST and 1 from GQN Mazes) with qualitative results, but not quantitative results on these benchmarks. It should be better to report some sort of quantitative comparisons and evaluated over the full dataset instead
>
> To address your comments, we have included 4 quantitative metrics (SSIM, PSNR, LPIPS, and FVD) on all datasets, including a new and substantially more complex Minecraft dataset (Guss et al., 2019), a short description of our experiments on which can be found in Section 4.4. Please also refer to Appendix D for additional detailed experimental results for all datasets and quantitative metrics.

---

### Author Response · Authors · 2020-11-25
**Authors' response**

We thank all reviewers for their constructive feedback!

We identified as the two main concerns (1) evaluation on a more challenging dataset and (2) reporting more comprehensive metrics.

To address the first point, we evaluated TALD on the Minecraft video sequences released by Guss et al. (2019). These contain videos of human players navigating to a goal location in a 3D procedurally generated world (traversing forests, mountains, villages, oceans, etc). We did not condition the models on the actions. Each model has trained on ~8000 training sequences of length 100 and evaluated multi-step predictions for 420 steps.

Please take a look at Figure 2, as well as Figure 15 in Appendix D, where we include 420-step open-loop video predictions for this dataset using four different models. TALD outperforms all baselines in terms of both quantitative metrics but also perceived prediction quality. TALD is the only model to achieve coherent multi-step predictions for 400 time steps, albeit being a bit blurry. In contrast, SVG fails to capture any long-range dependencies and VTA generates implausible events over long horizons.

To address the second point, we added Figure 12 in Appendix D to include 4 metrics (SSIM, PSNR, LPIPS, FVD) for all of our 4 datasets (Moving MNIST, KTH, GQN Mazes, Minecraft) and 4 models (TALD, VTA, RSSM, SVG).

We addressed the two main concerns, as well as smaller suggestions, in our updated paper and hope that this can lead to a revised evaluation of our submission. We respond to smaller suggestions and questions in our individual responses below.

---

### Decision · Program_Chairs · 2021-01-07
**Final Decision**

**Decision:**

Reject

**Comment:**

The paper studies the properties and advantages of temporal abstraction in hierarchical latent variable-based video prediction approaches, producing interesting results on various simulated environments and the KTH action dataset.

The two key drawbacks of this paper are: limited visual complexity of datasets used for evaluation (a real video dataset like BAIR pushing would have really helped), and lack of comparison (conceptually or empirically) to relevant prior work including those raised by various reviewers, plus see more below.

Aside from this, does the paper claim to propose a new model, or perform an empirical study to evaluate an existing model class, or both? The answer to this question is not always clear from the paper and this confusion is reflected both in the reviews and reviewer discussions and also in the authors' own responses to these.

Some potentially relevant works that don't find mention in this paper (aside from those pointed out by reviewers):
- NAOMI: Non-Autoregressive Multiresolution Sequence Imputation
https://arxiv.org/pdf/1901.10946.pdf
- Long-Horizon Visual Planning with Goal-Conditioned Hierarchical Predictors
https://arxiv.org/pdf/2006.13205.pdf

I do think these are all relatively easily fixed shortcomings, and I urge the authors to fix them in future versions.